# Fibrates ameliorate the course of bacterial sepsis by promoting neutrophil recruitment via CXCR2

Ivan Tancevski[1,*,†], Manfred Nairz[1,†], Kristina Duwensee[1,†], Kristina Auer[1,†], Andrea Schroll[1,†], Christiane Heim[1], Clemens Feistritzer[1], Julia Hoefer[2], Romana R Gerner[3], Alexander R Moschen[3], Ingrid Heller[4], Petra Pallweber[4], Xiaorong Li[5], Markus Theurl[6], Egon Demetz[1], Anna M Wolf[7], Dominik Wolf[7], Philipp Eller[8], Andreas Ritsch[3,†] & Guenter Weiss[1,†,**]

## Abstract

Bacterial sepsis results in high mortality rates, and new therapeutics to control infection are urgently needed. Here, we investigate the therapeutic potential of fibrates in the treatment of bacterial sepsis and examine their effects on innate immunity. Fibrates significantly improved the survival from sepsis in mice infected with *Salmonella typhimurium*, which was paralleled by markedly increased neutrophil influx to the site of infection resulting in rapid clearance of invading bacteria. As a consequence of fibrate-mediated early control of infection, the systemic inflammatory response was repressed in fibrate-treated mice. Mechanistically, we found that fibrates preserve chemotaxis of murine neutrophils by blocking LPS-induced phosphorylation of ERK. This results in a decrease of G protein-coupled receptor kinase-2 expression, thereby inhibiting the LPS-mediated downregulation of CXCR2, a chemokine receptor critical for neutrophil recruitment. Accordingly, application of a synthetic CXCR2 inhibitor completely abrogated the protective effects of fibrates in septicemia *in vivo*. Our results unravel a novel function of fibrates in innate immunity and host response to infection and suggest fibrates as a promising adjunct therapy in bacterial sepsis.

**Keywords** chemokine receptor; fenofibrate; infection; inflammation; neutrophils

**Subject Categories** Immunology; Microbiology, Virology & Host Pathogen Interaction

## Introduction

Sepsis is a systemic inflammatory condition following severe bacterial infection resulting in high mortality rates (Riedemann *et al*, 2003; Fauci *et al*, 2008). Sepsis is a global health problem with an increasing incidence over the past years, accounting for > 700,000 cases/year in the United States (Fauci *et al*, 2008). In sub-Saharan Africa, community-acquired bacteremia accounts for at least one-fourth of deaths in children above 1 year of age, with *Salmonella* species, *Streptococcus pneumoniae*, *Haemophilus influenzae*, and *Escherichia coli* being among the most commonly isolated pathogens (Fauci *et al*, 2008). Facing these high mortality rates and limited new developments of antibacterial drugs in an era of increasing anti-microbial resistance, novel treatment options for sepsis are urgently needed.

Fibrates have been used to treat hyperlipidemia for more than 40 years. However, their effects on cardiovascular outcome remain uncertain, restricting their use in the clinical routine (Jun *et al*, 2010; Goldfine *et al*, 2011). Two randomized, placebo-controlled trials of gemfibrozil demonstrated improvements in cardiovascular outcomes, but subsequent trials of bezafibrate and fenofibrate (the BIP, FIELD, and ACCORD studies) showed no significant overall cardiovascular benefit over placebo (Frick *et al*, 1987; Rubins *et al*, 1999; BIP, 2000; Keech *et al*, 2005; Ginsberg *et al*, 2010; Goldfine *et al*, 2011).

Fibrates are ligands of the peroxisome proliferator-activated receptor-alpha (PPARα), a member of the nuclear receptor superfamily of ligand-dependent transcription factors related to retinoid, steroid, and thyroid hormone receptors (Genovese *et al*, 2005). Previous studies indicated immune-regulatory properties of fibrates in macrophages and lymphocytes (Lee *et al*, 2007; Crisafulli & Cuzzocrea, 2009). Fibrates were shown to inhibit LPS/interferon-gamma (IFN-γ)-induced inflammation in macrophages, whereas they

1 Department of Internal Medicine VI/Infectious Diseases, Immunology, Rheumatology, Pneumology, Innsbruck Medical University, Innsbruck, Austria
2 Department of Urology, Innsbruck Medical University, Innsbruck, Austria
3 Department of Internal Medicine I/Gastroenterology, Endocrinology & Metabolism, Innsbruck Medical University, Innsbruck, Austria
4 Department of Hygiene and Medical Microbiology, Innsbruck Medical University, Innsbruck, Austria
5 Department of Pharmacology, Capital Medical University, Beijing, China
6 Department of Internal Medicine III/Cardiology, Innsbruck Medical University, Innsbruck, Austria
7 Department of Hematology/Oncology, University Hospital Bonn, Bonn, Germany
8 Department of Internal Medicine/Angiology, Medical University of Graz, Graz, Austria
   *Corresponding author. Tel: +43 512 50481602; Fax: +43 512 50425608; E-mail: ivan.tancevski@i-med.ac.at
   **Corresponding author. Tel: +43 512 50423251; Fax: +43 512 50425608; E-mail: guenter.weiss@i-med.ac.at
   †These authors contributed equally to this study.

failed to dampen the inflammatory response in macrophages from $Ppar\alpha^{-/-}$ mice (Crisafulli & Cuzzocrea, 2009). Moreover, PPARα activation was shown to inhibit the production of proinflammatory molecules by negatively interfering with the AP1 and NF-κB signaling pathway in classically activated macrophages (Marx *et al*, 2001; Neve *et al*, 2001; Rigamonti *et al*, 2008). In *interleukin (Il)-10$^{-/-}$* mice, fibrate treatment decreased the colonic expression of the pro-inflammatory cytokines IFN-γ and IL-17, which could be traced back to direct inhibition of IFN-γ and IL-17 expression in isolated T cells (Lee *et al*, 2007). However, the potential role of fibrates in bacterial infection has not been investigated so far.

Here, we found that fibrates promote neutrophil migration and early clearance of pathogens independently of PPARα, leading to significantly improved survival of septic mice. We found that fibrates preserve the expression of CXCR2 in neutrophils, a receptor crucially involved in chemokine sensing and migration. Finally, blockage of CXCR2 abolished enhanced influx of neutrophils to the site of infection and subsequently diminished the survival of fibrate-treated mice.

# Results

## Fenofibrate treatment increases the survival of septic mice

To study the impact of fibrate treatment on sepsis, we used a mouse model of *Salmonella* septicemia established in our laboratory (Nairz *et al*, 2009, 2011, 2013). In our first set of experiments, C57BL/6 mice were inoculated intraperitoneally (i.p.) with 50 colony-forming units (CFU) of *S. typhimurium* and were then set on a fenofibrate-supplemented (0.2% wt/wt) diet or control chow (control). Fenofibrate-treated mice showed a significantly increased survival rate compared to controls (Fig 1A). Fenofibrate-treated mice still displayed a significantly increased survival rate compared to controls receiving a standard diet, even when being inoculated with a 10-fold higher dose of *S. typhimurium* (500 CFU, Fig 1B). In line with an increased mortality rate, control mice presented with lethargy and piloerection as soon as 72 h after *S. typhimurium* injection, which are clinical signs of severe sepsis (Fig 1C). Because the study design with an inoculum of 500 CFU may rather resemble a severe clinical course of sepsis in humans, all further experiments were performed with this experimental setting.

To further elaborate on the mechanism underlying increased survival of fenofibrate-treated mice, we studied bacterial loads in liver and spleen after 3 days of infection showing markedly reduced bacterial numbers in fenofibrate-treated animals (Fig 1D and E). In parallel, the spleen weight, a surrogate for the severity of infection, was reduced by 40% in mice treated with fenofibrate (Fig 1F). Whereas fenofibrate-treated mice had a preserved splenic organ architecture, control animals presented with scattered inflammatory foci in the spleens due to the formation of multiple microabscesses (Fig 1I).

## Fenofibrate treatment represses the inflammatory response of septic mice

In line with the observation of reduced bacterial numbers in organs, fibrate-treated animals displayed significantly reduced serum concentrations of IL-6 (Fig 1G). Moreover, fenofibrate treatment significantly reduced mRNA expression of TNF-α, IL-6, the phagocyte oxidase subunit p47 (PHOX-p47), IL-12p35, IL-12p40, and IL-23p19, as well as expression of key T helper (Th) 17 and Th1 cell cytokines IL-17 and IFN-γ in spleens of *Salmonella*-infected mice on day 3 after initiation of infection (Fig 1H).

Fibrates were previously shown to repress NF-κB activation in macrophages, thereby dampening their pro-inflammatory response (Crisafulli & Cuzzocrea, 2009). To disentangle the mechanism underlying the repression of inflammation observed in the studies presented here, we injected mice intraperitoneally with a sublethal dose of LPS in the presence or absence of fibrates. While fenofibrate treatment significantly reduced the mRNA expression of TNF-α, no differences in splenic mRNA expression of IL-6, IL-12p40, PHOX-p47, IL-23p19, IL-17, and IFN-γ were observed as a consequence of fenofibrate supplementation (Supplementary Fig S1). Thus, the reduced expression of most cytokines in fibrate-treated mice suffering from *Salmonella* sepsis (Fig 1H) appeared to be the result of fibrate-mediated reduction of bacterial load, rather than being a consequence of direct effects of fibrates on cytokine formation.

Accordingly, the observed reduction in bacterial numbers may be traced back either to direct anti-microbial effects of fibrates or to a fibrate-triggered amelioration of innate immune responses. Using a disk diffusion test, we were able to exclude any direct bactericidal activity of fibrates toward *Salmonella* (Supplementary Fig S2).

## Fibrates promote neutrophil recruitment thereby enhancing early bacterial clearance

Successful early clearance of bacterial infection depends on efficient neutrophil migration to the site of infection (Alves-Filho *et al*, 2010). In a time-course experiment using an automated cell counter, we indeed found significantly increased neutrophil counts in the peritoneal cavity of fenofibrate-treated mice as soon as at 12 h after bacterial inoculation (Supplementary Fig S3), which was verified by May–Giemsa–Gruenwald staining in smears of peritoneal fluid (Fig 2A) and by Gr1$^+$ CD11b$^+$ F4/80$^-$ staining using FACS analysis (Supplementary Fig S4).

Accordingly, peritoneal leukocytes of fibrate-treated mice contained significantly more *S. typhimurium* (Fig 2B), whereas conversely, bacterial counts in peritoneal exudates of fenofibrate-treated mice were dramatically lower than in mice receiving a control diet (Fig 2C). Moreover, hepatic bacterial loads were markedly reduced in fibrate-treated mice at this early time-point (Fig 2D), demonstrating that fibrates prevent systemic invasion of *S. typhimurium* and sepsis by immediate clearance of bacteria from the site of infection. To further elucidate the mechanisms underlying rapid clearance of bacteria, we next studied the effect of fibric acids on uptake and killing of bacteria by neutrophils. As shown in Supplementary Figs S5–S7, fibrates did not modify the uptake of *S. typhimurium* by neutrophils, nor did they increase neutrophil ROS activity and degranulation. On the contrary, fibrates significantly inhibited neutrophil ROS activity and degranulation, which may protect from an overwhelming inflammatory response and associated end-organ failure *in vivo*.

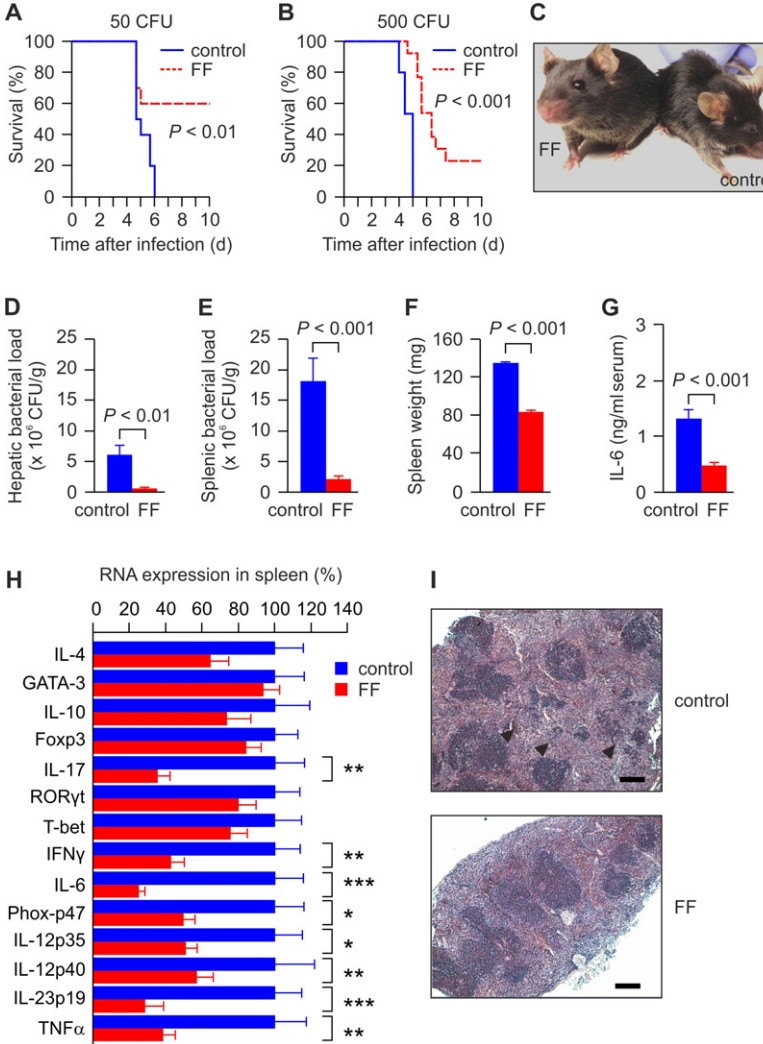

**Figure 1.  Fenofibrate increases the survival of mice with bacterial sepsis.**

Sepsis was induced in C57BL/6 mice by intraperitoneal injection of *Salmonella typhimurium*.

A    Survival rates of septic mice injected with 50 CFU of *S. typhimurium*, which were set on a fenofibrate (FF)—supplemented chow (0.2% wt/wt) or a standard diet without fibrate (control) immediately after injection of bacteria (*n* = 10).

B    Survival curve of septic mice injected with 500 CFU of *S. typhimurium* and set on a FF-supplemented chow (0.2% wt/wt) or a standard diet without fibrate (control) immediately after injection of bacteria (*n* = 19–20).

C    Clinical appearance of 0.2% FF-treated and control mice 72 h after inoculation with 500 CFU *S. typhimurium*.

D–H    Bacterial loads 72 h post-infection with 500 CFU in the liver (D) and spleen (E), as well as the spleen weight (F). Data are representative of four independent experiments, bars are means ± s.e.m. Seventy-two hours after intraperitoneal injection of 500 CFU of *S. typhimurium*, (G) serum concentrations of IL-6 were determined by ELISA, and (H) cytokine expression was determined in spleens of control and 0.2% FF-treated C57BL/6 mice by means of Taqman real-time PCR. Data are shown as means ± s.e.m.; *P < 0.05, **P < 0.01, ***P < 0.001 versus corresponding controls. Data are representative of two independent experiments (n = 14).

I    Spleens were further processed for HE staining. Whereas 0.2% FF-treated mice had preserved splenic organ architecture, control animals presented with scattered inflammatory foci in the spleens due to multiple microabscesses (arrow-heads). Scale bars represent 200 μm.

## The effect of fenofibrate on neutrophil influx and bacterial clearance is independent of PPARα

So far, most of the immune-modulatory effects of fibrates were shown to depend on the expression of the nuclear receptor PPARα, for which they serve as ligands (Marx *et al*, 2001; Neve *et al*, 2001; Genovese *et al*, 2005; Lee *et al*, 2007; Rigamonti *et al*, 2008; Crisafulli & Cuzzocrea, 2009). Thus, we next examined the effect of

fenofibrate treatment on *Salmonella* sepsis in *Pparα*$^{-/-}$ mice. Similarly as observed in C57BL/6 mice, *Pparα*$^{-/-}$ mice displayed significantly enhanced neutrophil influx to the site of infection 12 h after bacterial inoculation (Fig 3A) and reduced bacterial loads in liver and spleen after 3 days of infection when fed with fenofibrate as compared to mice receiving a control diet (Fig 3B and C). In parallel, the spleen weight was reduced by 35% in *Pparα*$^{-/-}$ mice treated with fenofibrate (Fig 3D). Thus, we conclude that fibrates improve

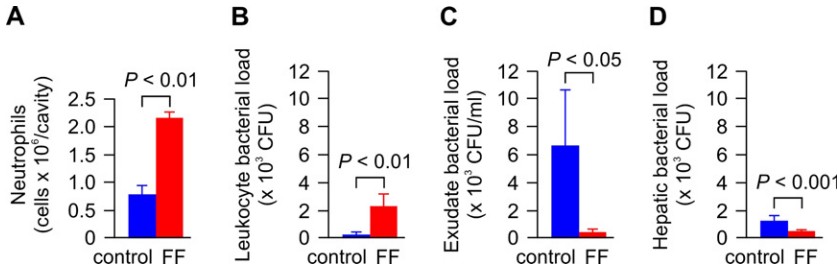

**Figure 2.  Fibrates increase neutrophil influx into the peritoneal cavity and promote early bacterial clearance.**

Septicemia was induced in C57BL/6 mice by i.p. injection of 500 CFU *Salmonella typhimurium*, and the animals were set on control or 0.2% FF-supplemented chow immediately after injection of bacteria.

A       After 12 h, number of intraperitoneal neutrophils was determined in smears of peritoneal fluid (*n* = 3).
B–D    Further, numbers of ingested bacteria in peritoneal leukocytes (B) and bacterial counts in the peritoneal exudates (C) and in livers (D) are shown (*n* = 7).

Data information: Data are means ± s.e.m., and are representative of three experiments.

the clinical course of systemic bacterial infection independently from PPARα.

### Fibrate-mediated neutrophil recruitment is not dependent on chemokine release from macrophages

The initiation of inflammatory responses is tightly regulated (Soehnlein & Lindbom, 2010; Mantovani *et al*, 2011). Owing to their strategic location in close proximity to the site of infection, tissue-resident macrophages are upon the primary inducers of an inflammatory reaction (Soehnlein & Lindbom, 2010; Mantovani *et al*, 2011) and as such promote the egress of neutrophils from the circulation by secreting chemokines including CXCL1 and CXCL2. We thus investigated whether the increased intraperitoneal accumulation of neutrophils in fenofibrate-treated mice was associated with an increased release of such chemoattractants.

Primary murine peritoneal macrophages treated with LPS showed a markedly increased secretion of CXCL1 and CXCL2, when compared to nonstimulated control cells. The addition of fibrates did not further increase the release of these chemokines. At high doses, fibrates even

decreased the release of CXCL1 and CXCL2 from activated macrophages over time (Supplementary Fig S8), which is in line with a previous observation indicating that fibrates dampen the inflammatory response in macrophages (Crisafulli & Cuzzocrea, 2009). Thus, we could rule out that an increased production of chemokines by resident macrophages is responsible for the enhanced attraction of neutrophils to the site of infection in fenofibrate-treated mice.

### Fibrates reverse LPS-driven loss of CXCR2 expression, a chemokine receptor crucial for neutrophil migration

The expression of CXCR2—the main chemokine receptor for CXCL1 and CXCL2—on neutrophils plays a critical role for their recruitment (Alves-Filho *et al*, 2009, 2010; Soehnlein & Lindbom, 2010). Downregulation of CXCR2 expression on circulating neutrophils is associated with impaired neutrophil migration to sites of infection during sepsis (Rios-Santos *et al*, 2007). The latter can be initiated via LPS-inducible TLR 4 activation, resulting in blockage of CXCR2 expression and neutrophil migration (Alves-Filho *et al*, 2010). Similarly, we found here that LPS downregulated CXCR2 expression on

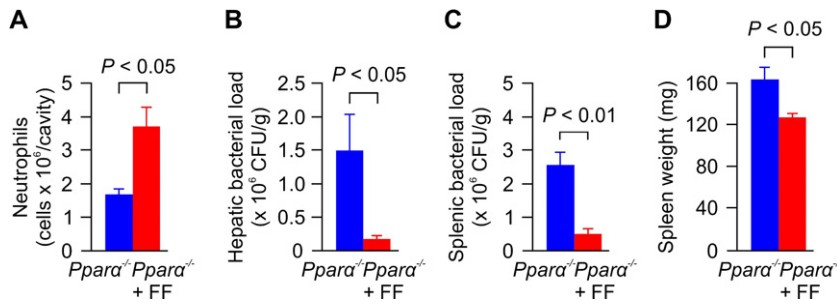

**Figure 3.  Fibrates increase neutrophil influx and bacterial clearance independently of PPARα.**

Sepsis was induced in *Pparα*^−/− mice by intraperitoneal injection of 500 CFU of *S. typhimurium*, and the animals were set on control or 0.2% FF-supplemented chow immediately after injection of bacteria.

A       Intraperitoneal neutrophil count 12 h after infection (*n* = 4).
B–D    Bacterial loads 72 h post-infection in the liver (B) and spleen (C), as well as the spleen weight (D) (*n* = 7).

Data information: Data are representative of 3 independent experiments, bars are means ± s.e.m.

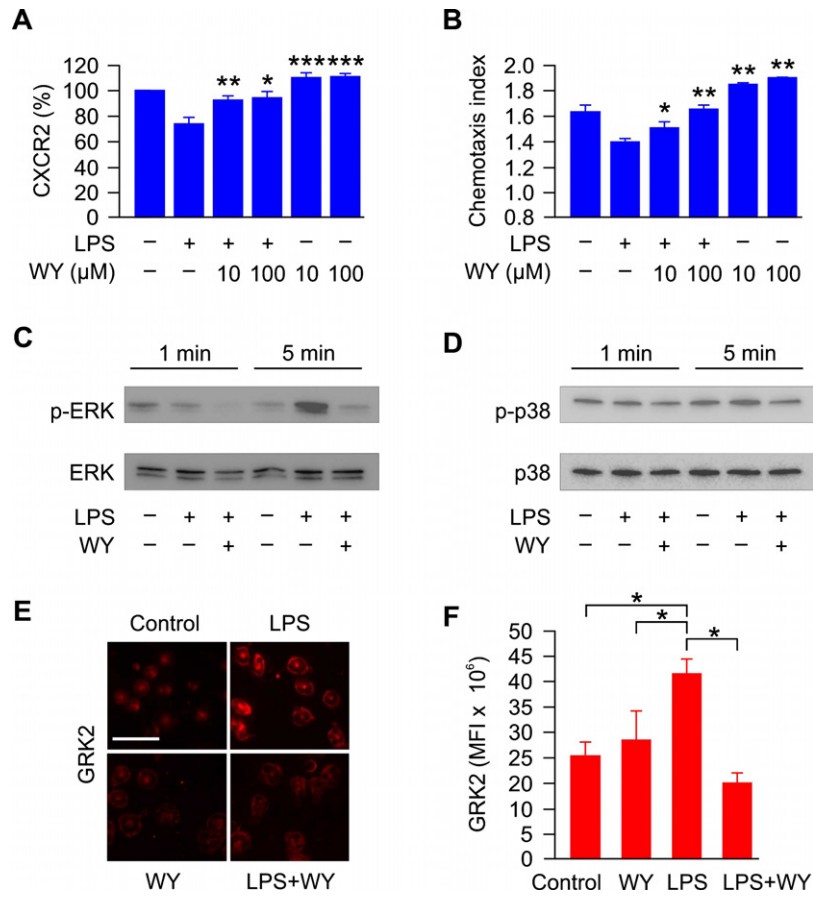

**Figure 4. Fibrates inhibit the downregulation of CXCR2 thereby preserving neutrophil functionality.**

A    The expression of CXCR2 was determined by FACS analysis of naive C57BL/6 bone marrow neutrophils cultured for 1 h with the fibrate WY-14643 (WY) or vehicle (DMSO), with LPS (10 µg/ml), or a combination of these reagents. Data are means ± s.e.m. from eight independent experiments; *$P < 0.05$, **$P < 0.01$ versus LPS.

B    Chemotaxis to CXCL2 (30 ng/ml) was determined in C57BL/6 bone marrow neutrophils pre-treated for 1 h with WY, or in combination with LPS (1 µg/ml), as indicated. Data are means ± s.e.m. from four independent experiments; *$P < 0.05$, **$P < 0.01$ versus LPS.

C, D    Representative Western blots showing phosphorylation of MAP kinases ERK (p-ERK) and p38 (p-p38) upon stimulation of murine neutrophils with vehicle, LPS (1 µg/ml), WY (10 µM), or a combination of these reagents for 1 and 5 min.

E    Representative immunofluorescence staining of GRK2 expression in fixed neutrophils purified from the bone marrow of naive C57BL/6 mice and cultured for 1 h with LPS (1 µg/ml), WY (10 µM), or a combination of both. Scale bars represent 25 µm.

F    Mean fluorescence intensity (MFI) of GRK2 in 3–4 visual fields per treatment group from experiment (E) was determined using ImageJ; *$P < 0.05$ versus LPS.

neutrophils, which was prevented by co-incubation with fibrates. This protective effect of fibrates was not only seen at a dose of 1 µg/ml LPS (unpublished data), but even at the higher dose of 10 µg/ml LPS (Fig 4A). In line with these findings, LPS-treated neutrophils had an impaired chemotactic response toward CXCL2 as compared to control cells, which was reversed upon concomitant treatment with fibrates (Fig 4B).

CXCR2 expression on neutrophiles is controlled by G protein-coupled receptor kinase-2 (GRK2), a TLR-ligand inducible serine-threonine protein kinase that causes the internalization of chemokine receptors (Penela *et al*, 2003; Vroon *et al*, 2006; Alves-Filho *et al*, 2009, 2010). This inflammation- and LPS-induced GRK2 activation can be traced back to phosphorylation of the mitogen-activated protein kinase (MAPK) ERK thereby blocking neutrophil chemotaxis (Ajibade *et al*, 2012; Liu *et al*, 2012). When investigating this pathway in isolated neutrophils, we found that fibrates block the LPS-driven phosphorylation of ERK (Fig 4C), and the associated

increase in GRK2 expression (Fig 4E and F). No significant effect of fibrates on p38 activation was observed (Fig 4D).

Moreover, we observed no effect of LPS and/or fibrates on neutrophil migratory efficiency toward fMLP and LTB4, other ligands important for neutrophil recruitment, pointing toward a major role for CXCR2 in fibrate–mediated promotion of chemotaxis (Supplementary Fig S9).

Importantly, fibric acids not only reversed LPS-mediated CXCR2 downregulation *in vitro*, but also preserved CXCR2 expression *in vivo* on circulating neutrophils of mice infected with *S. typhimurium*, corroborating the physiological relevance of our findings (Supplementary Fig S10).

These data indicate that fibrates prevent CXCR2 downregulation on neutrophils by inhibiting LPS-mediated ERK phosphorylation and the consecutive expression of GRK2, thereby preserving neutrophil functionality. This results in enhanced migration of neutrophils to the site of infection and promotes the clearance of pathogenic bacteria.

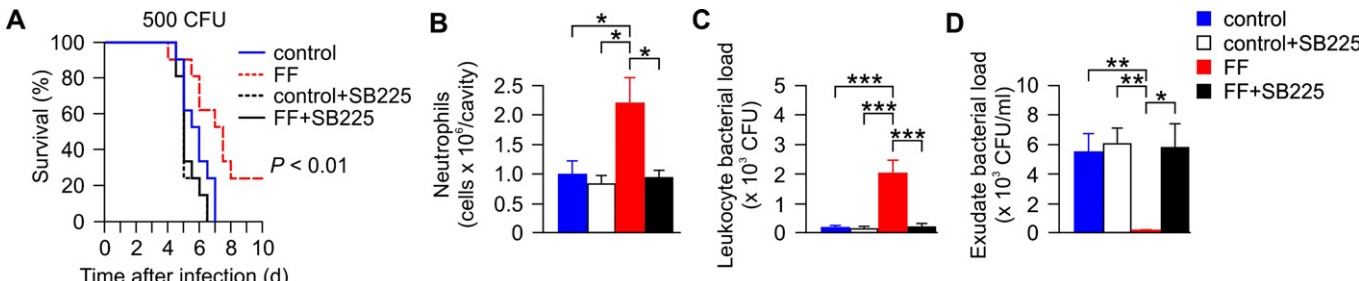

**Figure 5. Blockade of CXCR2 expression antagonizes the beneficial effects of fibrates in sepsis.**

A    Survival of septic mice intraperitoneally inoculated with 500 CFU of *Salmonella typhimurium* and set on a 0.2% FF-enriched or a standard chow (control) in the presence/absence of the CXCR2 inhibitor SB225002 (SB225, 10 mg/kg body weight) immediately after injection of bacteria; $n$ = 10, $P < 0.01$ for all groups versus FF.

B–D  In an independent, analogous experiment using SB225002, the number of intraperitoneal neutrophils (B), bacterial loads in peritoneal leukocytes (C), and within the peritoneal exudates (D) were determined after 12 h of infection as described in Fig 2. Data are means ± s.e.m., $n$ = 5; $*P < 0.05$, $**P < 0.01$, $***P < 0.001$ versus fenofibrate.

### Fibrate-mediated increase in neutrophil CXCR2 expression does not involve IL-33

Our results partly resembled effects which have been described for IL-33 toward induction of CXCR2 expression and neutrophil migration (Alves-Filho *et al*, 2010). We thus studied whether the novel effect of fibrates described herein may be traced back to induction of IL-33 and/or its soluble receptor ST2 (sST2) in neutrophils, thereby preventing CXCR2 downregulation via an autocrine loop.

However, IL-33 and sST2 formation in supernatants of murine neutrophils stimulated with LPS and treated with fibrates for up to 12 h were not significantly altered at any of the studied time-points with any of the treatments (Supplementary Fig S11A and B). Thus, an involvement of the IL-33 pathway in fibrate-mediated stabilization of CXCR2 expression in neutrophils appeared unlikely, and our results suggested a direct effect of fibrates on CXCR2-mediated amelioration of neutrophil function.

### Fibrate-mediated early bacterial clearance depends on neutrophil CXCR2 expression

Finally, we performed survival studies in *S. typhimurium*-infected mice, where CXCR2 expression was pharmacologically inhibited upon treatment with the specific, synthetic CXCR2 blocker SB225002. While fenofibrate treatment resulted in a significantly better survival compared to controls, application of SB225002 abrogated the protective effects of fenofibrate in sepsis and significantly reduced the survival of mice (Fig 5A).

In an independent series of experiments, we observed a significant increase in neutrophil accumulation in the peritoneal cavity of fenofibrate-treated mice which was associated with increased bacterial ingestion in peritoneal leukocytes and decreased bacterial numbers in peritoneal exudates (Fig 5B and D). In contrast, when fibrate-treated mice were concomitantly injected with SB225002, neutrophil counts, leukocyte- and exudate-associated bacterial loads were comparable to untreated controls (Fig 5B and D).

### Fibrates do not affect neutrophil-dependent sterile inflammation

Finally, we thought of investigating whether fibrates may not only promote chemotaxis of neutrophils, thereby leading to early bacterial clearance and inhibition of an overwhelming inflammatory response, but if they also affect the course of inflammation per se. In a first study, we showed that fibrates promote the migration of neutrophils only in the presence of high levels of LPS or upon exposure to bacteria, as we observed similar neutrophil numbers in peritoneal lavages of control and FF-treated mice 12 h after intraperitoneal injection of thioglycollate (Supplementary Fig S12). This observation was confirmed in another model of "sterile" inflammation by investigating the effects of fibrates in dextran sulfate sodium (DSS) induced colitis: We found that fibrates neither affected disease severity of DSS-colitis as reflected by loss of body weight and alterations of colon length (Supplementary Fig S13A and B), nor colonic expression of pro-inflammatory genes including IL-1β, IL-6, and TNF-α, nor did they significantly alter tissue damage as quantified by a histological colitis scoring system as compared to mice receiving a control diet (Supplementary Fig S13C–F). Taken together, our data convincingly show that the effect of fibrates is restricted to Gram-negative bacterial sepsis induced by infection with pathogens such as *S. typhimurium*.

## Discussion

We show here a novel function of fibrates in exerting protective effects during bacterial sepsis by promoting neutrophil recruitment and efficient early clearance of pathogenic bacteria at the site of infection which could be traced back to fibrate-mediated stabilization of CXCR2 expression. Previously, fibrates have been shown to mitigate inflammation in classically activated macrophages and to inhibit IL-17 and IFN-γ expression in isolated murine T cells (Cuzzocrea *et al*, 2004; Lee *et al*, 2007; Crisafulli & Cuzzocrea, 2009). However, in both macrophages as well as lymphocytes, the anti-inflammatory effects of fibrates critically depended on the presence of PPARα (Cuzzocrea *et al*, 2004; Lee *et al*, 2007; Crisafulli & Cuzzocrea, 2009). Importantly, the effects of fibrates on the preservation of neutrophil functionality and improved infection control were independent of PPARα expression, as confirmed by experiments using $Ppar\alpha^{-/-}$ mice.

Besides their well-known effects through activation of PPARα, fibrates have been shown to exert also PPARα-independent or "nongenomic" effects in hepatocytes, including the activation of

MAPKs (Rokos & Ledwith, 1997; Mounho & Thrall, 1999; Pauley *et al*, 2002), summarized in (Gardner *et al*, 2005). In leukocytes, MAPKs are involved in inflammation, apoptosis, and migration (Liu *et al*, 2012). Recently, it was shown that the MAPKs p38 and ERK have opposite effects on neutrophil migration: Whereas p38 promoted neutrophil migration, ERK inhibited it (Liu *et al*, 2012), which could be traced back to differential activation of GRK2, a crucial regulator of chemokine receptor expression (Liu *et al*, 2012).

From our results, it is conceivable that fibrates downregulate GRK2 expression in neutrophils by interfering with the phosphorylation of ERK. In the studies by Liu *et al*, activation of ERK and the associated potentiation of GRK2 expression resulted in internalization of formyl peptide-receptor 1 (FPR1), which is critical for directional movement of neutrophils toward chemoattractants such as bacterial N-formylpeptides (Liu *et al*, 2012). Another major receptor for neutrophil chemotaxis is CXCR2, which senses macrophage-derived chemokines (Soehnlein & Lindbom, 2010; Mantovani *et al*, 2011). In $Cxcr2^{-/-}$ mice, immunity against bacterial infection is severely compromised due to defective neutrophil chemotaxis (Tsai *et al*, 2000; Tateda *et al*, 2001). In addition, recent studies by Mei *et al* (2012) suggested CXCR2 to be a central regulator of neutrophil homeostasis under basal conditions. Both FPR1 and CXCR2 have been shown to be downregulated by induction of GRK2 in sepsis (Alves-Filho *et al*, 2009, 2010; Liu *et al*, 2012). Here, we provide compelling evidence that fibrates inhibit the LPS-inducible (p-ERK mediated) activation of GRK2 and the subsequent downregulation of CXCR2 in septic mice, thereby preserving neutrophil functionality and immune-cell-mediated pathogen elimination. In addition, selective blockage of CXCR2 *in vivo* inhibited the increased neutrophil transmigration to the site of infection in fenofibrate-treated mice. This suggested that fibrates act on neutrophil chemotaxis mainly by influencing CXCR2, without affecting FPR1 (summarized in Fig 6).

Although IL-33, a member of the IL-1 family, was the first molecule described to preserve CXCR2-mediated migration of neutrophils in sepsis (Alves-Filho *et al*, 2010), we found that the fibrate-mediated stabilization of CXCR2 occurred independently of this IL-33-mediated pathway.

Various formulations of fibrates have been used in clinical practice for decades and are generally well tolerated (Goldfine *et al*, 2011; Jackevicius *et al*, 2011). Here, we provide evidence for the first time that fibrates, routinely used to treat dyslipidemia in humans, may be a promising adjunct treatment to ameliorate bacterial clearance and to improve the outcome from sepsis, whereas due to the mechanism of action explored herein and supported by experimental data in thioglycollate-injected mice and DSS-colitis, fibrates do not affect the course of sterile inflammatory reactions. Given the long-lasting experience with this drug class, our results may readily be translated into the clinical setting. As a main consequence of early bacterial clearance by neutrophils, fibrate treatment resulted in a significant reduction of systemic pro-inflammatory cytokine levels. In addition, we found that fibrates drastically reduced the release of ROS from neutrophils which induce tissue damage at the site of infection. Inhibition of an overwhelming inflammatory response together with reduced tissue damage may have led to an improvement of undesirable hemodynamic changes, which in summary resulted in a significant better survival in mice with *Salmonella* sepsis receiving a fibrate-containing diet after the initiation of infection. Based on our results, we suggest that fibrates may

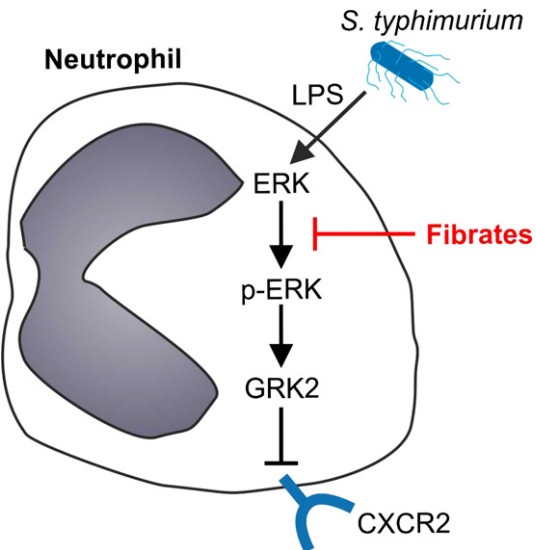

**Figure 6.  Suggested mechanisms underlying the protective effects of fibrates in sepsis.**
Our results indicate that fibrates block LPS-mediated activation of the MAPK-ERK and associated increase in GRK2 expression, and as a consequence inhibit degradation of CXCR2, thereby preserving chemotaxis and transmigration of murine neutrophils. By preserving CXCR2 expression on neutrophils, fibrates exert protective effects during bacterial sepsis by promoting rapid neutrophil migration to the site of infection, leading to an efficient early clearance of pathogens.

be a promising adjunct therapy (together with appropriate antibiotics and intensive care support) in the treatment of sepsis by promoting neutrophil influx to the primary site of infection, aiding in an efficient elimination of bacteria, and contributing to improved clinical outcomes from sepsis.

# Materials and Methods

### Chemicals and reagents

Fenofibrate and dextran sulfate sodium were purchased from Sigma-Aldrich (St. Louis, MO). For *in vitro* experiments, the fibrate WY-14643 (Sigma-Aldrich, St. Louis, MO) was used as described (Chinetti *et al*, 2000). LPS (from *Salmonella enterica* serovar *typhimurium*) was purchased from Sigma-Aldrich (St. Louis, MO), the CXCR2 antagonist SB225002 from Tocris Bioscience (Bristol, UK). Wild-type *Salmonella enterica* serovar *typhimurium* (*S. typhimurium*) strain ATCC14028 (American Type Culture Collection, Bethesda, MD) was used for all experiments and grown under sterile conditions in LB broth (Sigma-Aldrich, St. Louis, MO) to late-logarithmic phase (Nairz *et al*, 2011).

### Induction of *S. typhimurium* sepsis

All animal experiments were performed according to the guidelines of the Medical University of Innsbruck and the Austrian Ministry for Science and Education based on the Austrian Animal Testing Act of 1988 (BMWF-66.011/0012-2007-2010, BMWF-66.011/0059-II/3b/2013, and BMWF-66.011/0101-II/10b/2010). C57BL/6 and $Ppar\alpha^{-/-}$

mice (The Jackson Laboratory and Charles River, Sulzfeld, Germany) were housed under specific pathogen-free conditions at the Central animal facility of the Medical University of Innsbruck. For sepsis induction, male mice were used at 8–10 weeks of age and infected i.p. with indicated colony-forming units (CFU) of *S. typhimurium* diluted in 200 μl of PBS (Nairz *et al*, 2011). Where indicated, mice were injected i.p. with the CXCR2 antagonist SB225002 at 10 mg/kg 30 min before *Salmonella* inoculation (Alves-Filho *et al*, 2010). Immediately after injection of bacteria, mice were set on a standard chow diet or a chow diet supplemented with 0.2% fenofibrate (Ssniff, Soest, Germany), as indicated (Chinetti *et al*, 2000; Mardones *et al*, 2003). The animals were monitored twice daily for signs of illness for 10 days. In a subset of experiments, septic mice were sacrificed 72 h after induction of sepsis by cervical dislocation, and bacterial counts in organs were determined as described previously (Nairz *et al*, 2009, 2011, 2013).

### Induction of thioglycollate peritonitis

For induction of sterile inflammation, male C57BL/6 mice were used at 8–10 weeks of age and injected i.p. with 1 ml 4% thioglycollate (Sigma-Aldrich, St. Louis, MO). After 12 h, peritoneal leukocytes were collected by washing the peritoneal cavity with 10 ml PBS, and cell number was determined by the use of a XE 2100TM cell counter (Sysmex, Vienna, Austria).

### DSS-colitis

Dextran sulfate sodium (DSS)-colitis was induced in 8- to 10-week-old C57BL/6 mice essentially as described previously (Nairz *et al*, 2011). To induce acute colitis, mice were administered 3% DSS dissolved in drinking water for 5 days. DSS water was then replaced to autoclaved tap water until the end of experiment on day 8. Body weight, occult or gross blood, and stool consistency were assessed daily.

### Neutrophil influx

We collected peritoneal lavage fluids at indicated time-points after *S. typhimurium* infection. Total and differential cell counts were determined with the XE 2100TM cell counter (Sysmex, Vienna, Austria). Additionally, differential cell counts were carried out in smears stained with May-Gruenwald-Giemsa and using FACS analysis as described below. To determine *in vivo* phagocytosis in lavage fluids, leukocytes were collected by centrifugation, lyzed in 0.5% sodium deoxycholic acid (Sigma-Aldrich, St. Louis, MO) as previously described, and plated on LB agar plates (Sigma-Aldrich, St. Louis, MO) under sterile conditions in serial dilutions for 24 h (Nairz *et al*, 2011).

### Neutrophil isolation and chemotaxis assay

We isolated mouse neutrophils from naive C57BL/6 bone marrow by Ficoll density gradient. Chemotaxis in response to murine CXCL2 (30 ng/ml) (R&D Systems, Minneapolis, MN), LTB4 (100 nM) (Cayman Chemicals, Ann Arbor, MN), and fMLP (100 nM) (Sigma-Aldrich, St. Louis, MO) was performed using a modified 48-well Boyden microchemotaxis chamber (NeuroProbe, Gaithersburg, MD) as described (Feistritzer *et al*, 2006). Data are expressed as a chemotaxis index, which is the ratio between the distance of directed and random migration without attractants of neutrophils into the nitrocellulose filters (Feistritzer *et al*, 2006; Schroll *et al*, 2012).

### *In vitro* phagocytosis

Mouse neutrophils were isolated from naïve C57BL/6 bone marrow by Ficoll density gradient and diluted to a final concentration of $2 \times 10^6$ per well. Prior to infection, cells were incubated with vehicle (DMSO) or with the fibrate WY-14643 (Wy, 100 μM) for 30 min at 37°C. Then, neutrophils were infected with *S. typhimurium* at a multiplicity of infection (MOI) of 100 for 30 min at 37°C. Infected cells were washed with gentamicin to kill extracellular bacteria and harvested in 0.5% sodium deoxycholic acid. The lysates were plated onto LB agar plates and incubated at 37°C. CFUs were determined after 24 h.

### Neutrophil release of IL-33 and sST2

Freshly isolated murine neutrophils were incubated with vehicle, LPS (1 μg/ml), or a combination of LPS and the fibrate WY-14643 (WY, 10 μM) for 15, 30, 60 min, and 12 h. IL-33 and its soluble receptor ST2 (sST2) in cell culture supernatants were measured by ELISA. IL-6 measured by ELISA served to verify appropriate LPS stimulation of the cells.

### Macrophage isolation and stimulation with LPS

Resident peritoneal macrophages from naive mice were isolated as described (Nairz *et al*, 2011). Macrophages were cultured with vehicle, with LPS (1 μg/ml), or a combination of LPS with the fibrate WY-14643 (WY, 10 and 100 μM) for the indicated time-points. The secretion of chemoattractants CXCL1 and CXCL2 was determined by means of ELISA.

### Flow cytometry analysis

Cell surface staining was performed with antibodies against CXCR2 (242216, R&D Systems, Minneapolis, MN) (Alves-Filho *et al*, 2010), Gr1 (RB6-8C5, BD Biosciences, San Jose, CA) (Yang *et al*, 2002), CD11b, and F4/80 (both from BD Biosciences, San Jose, CA). Neutrophils were identified as (Gr1$^+$ CD11b$^+$ F4/80$^-$) (Swirski *et al*, 2009; Drechsler *et al*, 2010). Cells were analyzed on a Gallios Flow cytometer (Beckmann Coulter GmbH, Vienna, Austria). Neutrophil numbers were calculated as total cells multiplied by percent cells within the neutrophil gate (Swirski *et al*, 2009).

### Western blot

Freshly isolated murine neutrophils were incubated with vehicle, with LPS (1 μg/ml), or a combination of LPS and the fibrate WY-14643 (WY, 10 μM) for 1, and 5 min. Preparation of proteins and subsequent Western blot analysis were performed as described (Tancevski *et al*, 2010). Antibodies against ERK, p-ERK, p38, and p-p38 were from Cell Signaling (Danvers, MA). The chemoluminescent reaction was performed using Super Signal West Dura Reagent (Pierce, Rockford, IL), and blots were visualized by Fluor-S-Imager using Quantity One V4.1 software (Bio-Rad, Hercules, CA).

## Immunofluorescence staining

GRK2 staining and its quantification in murine neutrophils was performed similarly as described by Alves-Filho *et al* (2010). Bone marrow-derived murine neutrophils were seeded onto poly-D-lysine-coated glass coverslips and allowed to attach for 1 h in the presence of LPS (1 µg/ml), WY (10 µM), or a combination of both. Subsequently, cells were fixed with ice-cold methanol for 10 min. The cells were then washed with PBS and blocked with PBS/1% BSA for 30 min. Coverslips were incubated for 1 h with primary antibody against GRK2 (dilution 1:50; Abcam, Cambridge, UK), or isotype control (Dako, Glostrup, Denmark). Alexa-Fluor-555 goat anti-rabbit (1:250; Invitrogen, Carlsbad, CA) served as secondary antibody. The cells were visualized using fluorescent microscopy on a Zeiss Axio Imager M1 microscope (Zeiss, Oberkochen, Germany). Mean fluorescence intensity was determined by the use of ImageJ (http://rsbweb.nih.gov/ij/index.html).

## ELISA

IL-6, IL-33, sST2, CXCL1, and CXCL2 were measured by the use of Quantikine mouse ELISA kits (R&D Systems, Minneapolis, MN).

## RNA isolation, reverse transcription, and Taqman real-time PCR

Total RNA was extracted using RNA bee according to the manufacturer's protocol (Tel-test Inc) and reverse-transcribed with Omniscript-RT Kit (Qiagen, Hilden, Germany) (Tancevski *et al*, 2010). Primers and probes were described previously (Nairz *et al*, 2011). HPRT served as reference (Applied Biosystems, Foster City, CA). Taqman real-time PCRs were performed on a CFX96 PCR System (Bio-Rad, Hercules, CA).

## Measurement of neutrophil oxidative burst

Reactive oxygen species (ROS) production in neutrophils was determined by flow cytometry, using 2',7'-dichlorofluorescin diacetate (DCF-DA) (Sigma-Aldrich, St. Louis, MO). Freshly isolated primary murine neutrophils were pre-stimulated with vehicle (DMSO) or with the fibrate WY (100 µM) for 30 min at 37°C and then incubated simultaneously with DCF-DA (10 µM) and with heat-inactivated *S. typhimurium* at a multiplicity of infection (MOI) of 10 for 15 min at 37°C. Oxidative burst was measured immediately using flow cytometry (Gallios, Beckmann Coulter GmbH, Vienna, Austria). Results are expressed as mean fluorescence intensity.

## MPO assay in neutrophils

To determine myeloperoxidase activity in murine neutrophils, the MPO ELISA Kit NWK-MPO03 from Northwest Life Science Specialties, LLC (Vancouver, WA) was used according to the manufacturer's protocol.

## Statistical analysis

Statistical analysis was carried out using the SPSS statistical package (IBM, North Castle, NY). We determined significance by unpaired two-tailed Student's *t*-tests or by Mann–Whitney U-test to assess data where only two groups existed. Analysis of variance combined with Bonferroni correction or Kruskal–Wallis test, as appropriate, was used for all other experiments. Survival studies were analyzed with the log-rank test. $P < 0.05$ was considered statistically significant.

**Supplementary information** for this article is available online: http://embomolmed.embopress.org

## Acknowledgements

This work was supported by grants from the Austrian Research Fund, FWF (TRP-188, P19999-B05, and P23853-B13), and by the Medizinische Forschungsfoerderung Innsbruck (MFI No. 4316).

## Author contributions

IT, MN, KD, KA, AS, CH, CF, JH, RRG, ARM, IH, PP, XL, MT, ED, AMW, DW, AR, and PE conducted experiments and analyzed the data. IT and GW conceived the study, analyzed, and interpreted the data and wrote the manuscript.

## Conflict of interest

The authors declare that they have no conflict of interest.

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

### The paper explained

#### Problem
Sepsis is a global health problem with a mortality rate of 60–70%, accounting for almost 1,000,000 cases/year in industrialized countries. Facing these high mortality rates and limited new developments of antibacterial drugs in an era of increasing anti-microbial resistance, novel treatment options for sepsis are urgently needed.

#### Results
We show that fibrates, a drug class normally used to treat high blood lipids, protect from bacterial sepsis in mice by accelerating the recruitment of neutrophils to sites of infection. By boosting the migratory efficiency of these highly specialized white blood cells, fibrates promote early clearance of bacteria, thereby drastically reducing sepsis-driven mortality.

#### Impact
This study may change the treatment of sepsis in humans. Based on our results, we suggest that fibrates may be a promising adjunct therapy in the treatment of sepsis by promoting efficient elimination of bacteria and contributing to improved clinical outcomes from sepsis.

Alves-Filho JC, Sonego F, Souto FO, Freitas A, Verri WA Jr, Auxiliadora-Martins M, Basile-Filho A, McKenzie AN, Xu D, Cunha FQ *et al* (2010) Interleukin-33 attenuates sepsis by enhancing neutrophil influx to the site of infection. *Nat Med* 16: 708–712

BIP (2000) Secondary prevention by raising HDL cholesterol and reducing triglycerides in patients with coronary artery disease: the Bezafibrate Infarction Prevention (BIP) study. *Circulation* 102: 21–27

Chinetti G, Gbaguidi FG, Griglio S, Mallat Z, Antonucci M, Poulain P, Chapman J, Fruchart JC, Tedgui A, Najib-Fruchart J *et al* (2000) CLA-1/SR-BI is expressed in atherosclerotic lesion macrophages and regulated by activators of peroxisome proliferator-activated receptors. *Circulation* 101: 2411–2417

Crisafulli C, Cuzzocrea S (2009) The role of endogenous and exogenous ligands for the peroxisome proliferator-activated receptor alpha (PPAR-alpha) in the regulation of inflammation in macrophages. *Shock* 32: 62–73

Cuzzocrea S, Di Paola R, Mazzon E, Genovese T, Muia C, Centorrino T, Caputi AP (2004) Role of endogenous and exogenous ligands for the peroxisome proliferators activated receptors alpha (PPAR-alpha) in the development of inflammatory bowel disease in mice. *Lab Invest* 84: 1643–1654

Drechsler M, Megens RT, van Zandvoort M, Weber C, Soehnlein O (2010) Hyperlipidemia-triggered neutrophilia promotes early atherosclerosis. *Circulation* 122: 1837–1845

Fauci AS, Braunwald E, Kasper DL, Hauser SL, Longo DL, Jameson JL, Loscalzo J (2008) *Harrison's Principles of Internal Medicine*, 17th edn. New York: McGraw-Hill

Feistritzer C, Mosheimer BA, Sturn DH, Riewald M, Patsch JR, Wiedermann CJ (2006) Endothelial protein C receptor-dependent inhibition of migration of human lymphocytes by protein C involves epidermal growth factor receptor. *J Immunol* 176: 1019–1025

Frick MH, Elo O, Haapa K, Heinonen OP, Heinsalmi P, Helo P, Huttunen JK, Kaitaniemi P, Koskinen P, Manninen V *et al* (1987) Helsinki Heart Study: primary-prevention trial with gemfibrozil in middle-aged men with dyslipidemia. Safety of treatment, changes in risk factors, and incidence of coronary heart disease. *N Engl J Med* 317: 1237–1245

Gardner OS, Dewar BJ, Graves LM (2005) Activation of mitogen-activated protein kinases by peroxisome proliferator-activated receptor ligands: an example of nongenomic signaling. *Mol Pharmacol* 68: 933–941

Genovese T, Mazzon E, Di Paola R, Muia C, Crisafulli C, Caputi AP, Cuzzocrea S (2005) Role of endogenous and exogenous ligands for the peroxisome proliferator-activated receptor alpha in the development of bleomycin-induced lung injury. *Shock* 24: 547–555

Ginsberg HN, Elam MB, Lovato LC, Crouse JR III, Leiter LA, Linz P, Friedewald WT, Buse JB, Gerstein HC, Probstfield J *et al* (2010) Effects of combination lipid therapy in type 2 diabetes mellitus. *N Engl J Med* 362: 1563–1574

Goldfine AB, Kaul S, Hiatt WR (2011) Fibrates in the treatment of dyslipidemias–time for a reassessment. *N Engl J Med* 365: 481–484

Jackevicius CA, Tu JV, Ross JS, Ko DT, Carreon D, Krumholz HM (2011) Use of fibrates in the United States and Canada. *JAMA* 305: 1217–1224

Jun M, Foote C, Lv J, Neal B, Patel A, Nicholls SJ, Grobbee DE, Cass A, Chalmers J, Perkovic V (2010) Effects of fibrates on cardiovascular outcomes: a systematic review and meta-analysis. *Lancet* 375: 1875–1884

Keech A, Simes RJ, Barter P, Best J, Scott R, Taskinen MR, Forder P, Pillai A, Davis T, Glasziou P *et al* (2005) Effects of long-term fenofibrate therapy on cardiovascular events in 9795 people with type 2 diabetes mellitus (the FIELD study): randomised controlled trial. *Lancet* 366: 1849–1861

Lee JW, Bajwa PJ, Carson MJ, Jeske DR, Cong Y, Elson CO, Lytle C, Straus DS (2007) Fenofibrate represses interleukin-17 and interferon-gamma expression and improves colitis in interleukin-10-deficient mice. *Gastroenterology* 133: 108–123

Liu X, Ma B, Malik AB, Tang H, Yang T, Sun B, Wang G, Minshall RD, Li Y, Zhao Y *et al* (2012) Bidirectional regulation of neutrophil migration by mitogen-activated protein kinases. *Nat Immunol* 13: 457–464

Mantovani A, Cassatella MA, Costantini C, Jaillon S (2011) Neutrophils in the activation and regulation of innate and adaptive immunity. *Nat Rev Immunol* 11: 519–531

Mardones P, Pilon A, Bouly M, Duran D, Nishimoto T, Arai H, Kozarsky KF, Altayo M, Miquel JF, Luc G *et al* (2003) Fibrates down-regulate hepatic scavenger receptor class B type I protein expression in mice. *J Biol Chem* 278: 7884–7890

Marx N, Mackman N, Schonbeck U, Yilmaz N, Hombach V, Libby P, Plutzky J (2001) PPARalpha activators inhibit tissue factor expression and activity in human monocytes. *Circulation* 103: 213–219

Mei J, Liu Y, Dai N, Hoffmann C, Hudock KM, Zhang P, Guttentag SH, Kolls JK, Oliver PM, Bushman FD *et al* (2012) Cxcr2 and Cxcl5 regulate the IL-17/G-CSF axis and neutrophil homeostasis in mice. *J Clin Invest* 122: 974–986

Mounho BJ, Thrall BD (1999) The extracellular signal-regulated kinase pathway contributes to mitogenic and antiapoptotic effects of peroxisome proliferators in vitro. *Toxicol Appl Pharmacol* 159: 125–133

Nairz M, Schleicher U, Schroll A, Sonnweber T, Theurl I, Ludwiczek S, Talasz H, Brandacher G, Moser PL, Muckenthaler MU *et al* (2013) Nitric oxide-mediated regulation of ferroportin-1 controls macrophage iron homeostasis and immune function in Salmonella infection. *J Exp Med* 210: 855–873

Nairz M, Schroll A, Moschen AR, Sonnweber T, Theurl M, Theurl I, Taub N, Jamnig C, Neurauter D, Huber LA *et al* (2011) Erythropoietin contrastingly affects bacterial infection and experimental colitis by inhibiting nuclear factor-kappaB-inducible immune pathways. *Immunity* 34: 61–74

Nairz M, Theurl I, Schroll A, Theurl M, Fritsche G, Lindner E, Seifert M, Crouch ML, Hantke K, Akira S *et al* (2009) Absence of functional Hfe protects mice from invasive Salmonella enterica serovar typhimurium infection via induction of lipocalin-2. *Blood* 114: 3642–3651

Neve BP, Corseaux D, Chinetti G, Zawadzki C, Fruchart JC, Duriez P, Staels B, Jude B (2001) PPARalpha agonists inhibit tissue factor expression in human monocytes and macrophages. *Circulation* 103: 207–212

Pauley CJ, Ledwith BJ, Kaplanski C (2002) Peroxisome proliferators activate growth regulatory pathways largely via peroxisome proliferator-activated receptor alpha-independent mechanisms. *Cell Signal* 14: 351–358

Penela P, Ribas C, Mayor F Jr (2003) Mechanisms of regulation of the expression and function of G protein-coupled receptor kinases. *Cell Signal* 15: 973–981

Riedemann NC, Guo RF, Ward PA (2003) Novel strategies for the treatment of sepsis. *Nat Med* 9: 517–524

Rigamonti E, Chinetti-Gbaguidi G, Staels B (2008) Regulation of macrophage functions by PPAR-alpha, PPAR-gamma, and LXRs in mice and men. *Arterioscler Thromb Vasc Biol* 28: 1050–1059

Rios-Santos F, Alves-Filho JC, Souto FO, Spiller F, Freitas A, Lotufo CM, Soares MB, Dos Santos RR, Teixeira MM, Cunha FQ (2007) Down-regulation of CXCR2 on neutrophils in severe sepsis is mediated by inducible nitric oxide synthase-derived nitric oxide. *Am J Respir Crit Care Med* 175: 490–497

Rokos CL, Ledwith BJ (1997) Peroxisome proliferators activate extracellular signal-regulated kinases in immortalized mouse liver cells. *J Biol Chem* 272: 13452–13457

Rubins HB, Robins SJ, Collins D, Fye CL, Anderson JW, Elam MB, Faas FH, Linares E, Schaefer EJ, Schectman G *et al* (1999) Gemfibrozil for the

secondary prevention of coronary heart disease in men with low levels of high-density lipoprotein cholesterol. Veterans Affairs High-Density Lipoprotein Cholesterol Intervention Trial Study Group. *N Engl J Med* 341: 410−418

Schroll A, Eller K, Feistritzer C, Nairz M, Sonnweber T, Moser PA, Rosenkranz AR, Theurl I, Weiss G (2012) Lipocalin-2 ameliorates granulocyte functionality. *Eur J Immunol* 42: 3346−3357

Soehnlein O, Lindbom L (2010) Phagocyte partnership during the onset and resolution of inflammation. *Nat Rev Immunol* 10: 427−439

Swirski FK, Nahrendorf M, Etzrodt M, Wildgruber M, Cortez-Retamozo V, Panizzi P, Figueiredo JL, Kohler RH, Chudnovskiy A, Waterman P *et al* (2009) Identification of splenic reservoir monocytes and their deployment to inflammatory sites. *Science* 325: 612−616

Tancevski I, Demetz E, Eller P, Duwensee K, Hoefer J, Heim C, Stanzl U, Wehinger A, Auer K, Karer R *et al* (2010) The liver-selective thyromimetic T-0681 influences reverse cholesterol transport and atherosclerosis development in mice. *PLoS ONE* 5: e8722

Tateda K, Moore TA, Newstead MW, Tsai WC, Zeng X, Deng JC, Chen G, Reddy R, Yamaguchi K, Standiford TJ (2001) Chemokine-dependent neutrophil

recruitment in a murine model of Legionella pneumonia: potential role of neutrophils as immunoregulatory cells. *Infect Immun* 69: 2017−2024

Tsai WC, Strieter RM, Mehrad B, Newstead MW, Zeng X, Standiford TJ (2000) CXC chemokine receptor CXCR2 is essential for protective innate host response in murine Pseudomonas aeruginosa pneumonia. *Infect Immun* 68: 4289−4296

Vroon A, Heijnen CJ, Kavelaars A (2006) GRKs and arrestins: regulators of migration and inflammation. *J Leukoc Biol* 80: 1214−1221

Yang KK, Dorner BG, Merkel U, Ryffel B, Schutt C, Golenbock D, Freeman MW, Jack RS (2002) Neutrophil influx in response to a peritoneal infection with Salmonella is delayed in lipopolysaccharide-binding protein or CD14-deficient mice. *J Immunol* 169: 4475−4480

