## [Review Process File · EMBO Molecular Medicine]

Fibrates ameliorate the course of bacterial sepsis by promoting neutrophil recruitment via CXCR2

Ivan Tancevski, Manfred Nairz, Kristina Duwensee, Kristina Auer, Andrea Schroll, Christiane Heim, Clemens Feistritzer, Julia Hoefler, Romana R. Gerner, Alexander R. Moschen, Ingrid Heller, Petra Pallweber, Xiaorong Li, Markus Theurl, Egon Demetz, Anna M. Wolf, Dominik Wolf, Philipp Eller, Andreas Ritsch, Guenter Weiss

Corresponding author: Ivan Tancevski, Innsbruck Medical University

Review timeline:

Submission date:	21 August 2013
Editorial Decision:	01 October 2013
Revision received:	04 February 2014
Editorial Decision:	21 February 2014
Revision received:	07 March 2014
Accepted:	11 March 2014

Transaction Report:

Editor: Céline Carret

1st Editorial Decision

01 October 2013

Thank you for the submission of your manuscript to EMBO Molecular Medicine. I am sorry that it has taken so long to get back to you on your manuscript.

While reviewers 1 and 3 delivered their evaluations in a timely manner, we did not receive the other reviewers' input. As the evaluations from the first two reviewers are consistent, and a further delay cannot be justified, I have decided to proceed based on these evaluations. If in the meanwhile we should receive the other review within the next 10 days and only if it raises significant caveats, these would need to be taken into consideration. We would not, however, ask you to comply with any further-reaching requests.

You will see that while both reviewers are generally supportive of your work and underline its potential interest, they also raise a number of specific concerns that must be addressed in a major revision of the study.

As you will see from the comments below, both referees have the same main concern regarding the mechanistic insights and suggest further experiments to validate it by exploring possible alternative mechanisms. In addition, a number of questions regarding the effect of fibrates in the *in vivo* model are asked, that need to be answered to strengthen the main conclusions and clinical impact of the

findings.

In our view the suggested revisions would render the manuscript much more compelling and interesting to a broad readership. We therefore hope that you will be prepared to undertake the recommended experimental revision.

Please note that it is EMBO Molecular Medicine policy to allow only a single round of revision and that, as acceptance or rejection of the manuscript will depend on another round of review, your responses should be as complete as possible.

***** Reviewer's comments *****

Referee #1 (Comments on Novelty/Model System):

The authors report the interesting and possibly clinically relevant observation that fibrates partially protect mice from a lethal dose of Salmonella sepsis. They provide one likely explanation, i. e. the inhibition of LPS-induced downregulation of the chemokine receptor CXCR2 on neutrophils. While the data support that explanation, I miss a detailed analysis of potential alternative mechanisms (see below for further details).

Referee #1 (Remarks):

In this manuscript, Tancevski et al. describe a novel effect of fibrates, namely that they partially protect mice from lethal sepsis triggered by intraperitoneal injection of Salmonella bacteria. This observation correlates with increased peritoneal accumulation of neutrophils and increased bacterial uptake by leukocytes. However, this effect is not mediated by PPARalpha. The authors propose that fibrates prevent the LPS-induced downregulation of CXCR2, leading to higher level of neutrophil influx into the inflamed peritoneum.

The basic observations are interesting and novel, and may have human clinical relevance. The experiments are well designed and performed and mostly support the authors' conclusions. The proposed mechanism is likely at least partially responsible for the observed phenotype. However, there are a number of other potential mechanisms that the authors should address to strengthen the conclusion of the manuscript.

Specific points:

- 1) The proposed CXCR2-mediated mechanism may be partially responsible for the observed in vivo phenotypes. However, the effect of fibrates on CXCR2 expression and in vitro migration is much less pronounced than the changes in in vivo migration and bacterial clearance. Therefore, the authors should address possible alternative explanations/mechanisms. How do fibrates affect bacterial uptake and killing by neutrophils? How are responses to other GPCR ligands (fMLP, LTB4) affected? What about functional responses other than migratin (e. g. respiratory burst, degranulation, etc.)?
- 2) Is the effect of fibrates restricted to bacterial sepsis or do they also augment other in vivo responses that rely on neutrophil migration/CXCR2? Possible neutrophil-mediated sterile inflammatory responses may include thioglycollate peritonitis, the Arthus reaction of K/BxN serum-transfer arthritis. The effect of fibrates on those responses should be tested.
- 3) How do fibrates affect polymicrobial sepsis, e. g. in a cecal ligation and puncture model?

- 4) Since fibrates have been around for a while, I am surprised that there is no clinical data on their effect in sepsis. The authors should dig deeper into the literature and maybe attempt to perform a meta-analysis of the published human clinical data.
- 5) How do fibrates affect sepsis survival in PPARalpha knockout mice?

Referee #3 (Remarks):

In the present study Tancevski et. al. investigate the potential protective actions of fibrates in a murine model of bacterial induced sepsis. The authors provide evidence that administration of fenofibrate leads to protection from *Salmonella typhimurium* induced sepsis. They also demonstrate that PPAR α , the receptor described to mediate the cardiovascular actions of fibrates is not involved in this mechanism, whereas CXCR2 is key in mediating the protective actions of fenofibrate in vivo and WY-1463 in vitro. The experiments appear to be well conducted and the overall concept is very intriguing and could have potential clinical impacts. However there are several points that need to be addressed. They are as follows:

- 1) The actions demonstrated for WY-1463 are in the high micromolar range, what are the circulating levels of fenofibrate that are reached via dietary supplementation. In addition how do these levels correlate with the some the toxic effects associated with fibrates?
- 2) The authors demonstrate that PPAR α is not required for the protective actions displayed by fenofibrate in vivo, is this true also for the actions demonstrate for WY-1463 in vitro? The authors also need to discuss the rationale for using of a different compound for their in vitro studies to that employed for the in vivo study.
- 3) The authors show that WY-1463 inhibits the down regulation of CXCR2 by LPS and preserves PMN chemotactic abilities. In addition they also demonstrate that in vivo fenofibrate increases early PMN recruitment to the site of inflammation and that inhibition of CXCR2 in vivo reverses these protective actions. Since PPAR α is not involved in these protective actions are these actions mediated by another receptor or do the authors propose that these fibrates may directly regulate ERK and GRK2 phosphorylation?
- 4) It is note worthy that fenofibrates also increase bacterial phagocytosis in vivo, do they also increase bacterial killing either by stimulating ROS production or by upregulating other bactericidal mechanisms such as bacterial peptides and NET formation. In addition the authors need to provide detailed information on how the in vivo bacterial phagocytosis was evaluated in these experiments since no information is give in the manuscript in this regard.
- 5) Is CXCR2 expression in vivo also preserved in mice treated with fenofibrates?
- 6) On page 11 the authors state that ' This suggested that fibrates act on neutrophil chemotaxis mainly by influencing CXCR2, without affecting FPR1'. However no evidence is provided that FPR1 ligands are important in the in vivo experiments for recruitment of neutrophils to the site of inflammation. In addition the authors need to conduct side-by-side experiments in vitro with both FPR1 agonists (such as fMLP) and CXCR2 agonists to determine if in fact the reduction in receptor phosphorylation by GRK2 is restricted to CXCR2.
- 7) What is the influence of fibrates on the recruitment of other phagocytes to the site of inflammation such as monocytes?
- 8) It is note worthy that the message for a number of Th1 cytokines, such as IFN γ , which have been suggested to exert protective actions in sepsis are downregulated. Is this regulation observed at the protein level too? Also a discussion is merited for this apparent discrepancy between the current finding and those published in the literature.
- 9) The authors on page 5 state that spleen weight is a surrogate for the severity of infection, a reference for this statement should be provided.
- 10) The authors employ IP injection of LPS to determine the actions of fenofibrate on splenic cytokine production, this approach assumes that bacteria do not escape the peritoneal cavity to directly elicit humoral responses in the vasculature and even in the spleen. Thus the authors should determine whether their results with i.p. administration of LPS also hold true when LPS is administered i.v.
- 11) In a number of places in the results section (such as on page 7) the authors state that the mice were either treated with fenofibrate or solvent? What solvent does this statement refer to? In the materials and methods the authors state that the treatment was provided in the diet and not administered directly via injection.

Reviewer's comments

Referee #1 (Comments on Novelty/Model System):

The authors report the interesting and possibly clinically relevant observation that fibrates partially protect mice from a lethal dose of Salmonella sepsis. They provide one likely explanation, i. e. the inhibition of LPS-induced down regulation of the chemokine receptor CXCR2 on neutrophils. While the data support that explanation, I miss a detailed analysis of potential alternative mechanisms (see below for further details).

Referee #1 (Remarks):

In this manuscript, Tancevski et al. describe a novel effect of fibrates, namely that they partially protect mice from lethal sepsis triggered by intraperitoneal injection of Salmonella bacteria. This observation correlates with increased peritoneal accumulation of neutrophils and increased bacterial uptake by leukocytes. However, this effect is not mediated by PPARalpha. The authors propose that fibrates prevent the LPS-induced down regulation of CXCR2, leading to higher level of neutrophil influx into the inflamed peritoneum.

The basic observations are interesting and novel, and may have human clinical relevance. The experiments are well designed and performed and mostly support the authors' conclusions. The proposed mechanism is likely at least partially responsible for the observed phenotype. However, there are a number of other potential mechanisms that the authors should address to strengthen the conclusion of the manuscript.

Specific points:

1) The proposed CXCR2-mediated mechanism may be partially responsible for the observed in vivo phenotypes. However, the effect of fibrates on CXCR2 expression and in vitro migration is much less pronounced than the changes in in vivo migration and bacterial clearance. Therefore, the authors should address possible alternative explanations/mechanisms.

- *How do fibrates affect bacterial **uptake** and **killing by neutrophils**?*
- *How are responses to other GPCR ligands (**fMLP**, **LTB4**) affected?*
- *What about functional responses other than migration (e. g. **respiratory burst**, **degranulation**, etc.)?*

We thank the Reviewer for these suggestions. As suggested by the Reviewer we performed a number of new experiments addressing these questions. The results of these experiments critically improved the quality of our manuscript by elucidating the effect of fibrates on neutrophil bacterial uptake, killing (ROS, MPO), and migration towards other ligands including fMLP and LTB₄. As shown below, and now incorporated into the revised manuscript as Supplementary files, fibrates did not affect the uptake of *S. typhimurium* into neutrophils (**Figure S5**). In order to evaluate a possible effect of fibrates on anti-bacterial effector mechanisms of neutrophils, we examined the formation of ROS by neutrophils by performing a DCF assay and found that fibrates rather decreased ROS formation (**Figure S6**). Moreover, fibrates markedly decreased degranulation of neutrophils, as shown in **Figure S7**. In addition, and following the suggestions of both Reviewers 1 and 2, the chemotactic activity of neutrophils against different chemo-attractants including fMLP and LTB₄ was performed (**Figure S9**). Thereby, we found clear evidence that both LPS as well as fibrates influenced only the migration of neutrophils towards the specific CXCR2 ligand CXCL2 (**Figure S9** and **Figure 4B**), while neither LPS nor fibrates affected the migratory properties of neutrophils towards fMLP and LTB₄. Taken together, our results show that fibrates promote the migratory efficiency of neutrophils in mice infected with the Gram negative bacterium *S. typhimurium*, leading to early clearance of infection without affecting their phagocytic properties. Our additional experiments further confirm that this effect is specifically mediated by fibrate mediated antagonization of LPS-mediated de-stabilization of CXCR2 expression. Concomitantly, fibrates

appear to dampen the killing activity of murine neutrophils, which *in vivo* may protect from an overwhelming inflammatory response and associated end-organ failure.

Figure S5

Figure S5. Fibrates do not affect neutrophil bacterial uptake.

Freshly isolated murine neutrophils were incubated with vehicle (DMSO) or the fibrate WY-14643 (WY, 10 μ M) for 30 min. Then, neutrophils were infected with *S. typhimurium* at a MOI of 100 for 30 min at 37°C. Infected cells were washed with gentamicin to kill extracellular bacteria and harvested in 0.5% sodium deoxycholic acid. The lysates were plated onto LB agar plates and incubated at 37°C. CFUs were determined after 24 h. Data are means \pm SEM, n = 7.

Figure S6

Figure S6.

Fibrates decrease neutrophil ROS activity.

Reactive oxygen species (ROS) production was determined by flow cytometry, using 2', 7'-Dichlorofluorescein diacetate. Freshly isolated murine neutrophils were pre-stimulated with vehicle or with the fibrate WY (100 μ M) for 30 min at 37°C, and then incubated with heat-inactivated *S. typhimurium* at a MOI of 10 for 15 min. Results are expressed as mean fluorescence intensity, the histogram shows means for control and WY-treated cells, n = 5.

Figure S7

Figure S7. Fibrates decrease neutrophil MPO activity.

Myeloperoxidase (MPO) activity was measured in freshly isolated murine neutrophils pre-stimulated with vehicle (DMSO) or with the fibrate WY (100 μ M) for 30 min at 37°C. Data are means \pm SEM, n = 6.

Figure S9**Figure S9. Fibrates selectively improve CXCR2-mediated chemotaxis of neutrophils.**

Chemotaxis to LTB₄ (100 nM), fMLP (100 nM) and CXCL2 (30 ng/ml) was determined in C57BL/6 bone marrow neutrophils pre-treated for 1 h with WY (10 μM), or in combination with LPS (1 μg/ml), as indicated. Data are pooled from 2 independent experiments.

2) Is the effect of fibrates restricted to bacterial sepsis or do they also augment other in vivo responses that rely on neutrophil migration/CXCR2? Possible neutrophil-mediated sterile inflammatory responses may include **thioglycollate peritonitis**, the Arthus reaction of K/BxN serum-transfer arthritis. The effect of fibrates on those responses should be tested.

We would like to thank the Reviewer for this interesting question!

Following the Reviewer's suggestion, we tested the effects of fibrates in two non-infectious models of inflammation ("sterile inflammation"), namely in thioglycollate-induced peritonitis, and in dextran sulfate sodium (DSS)-induced colitis. In both models, fenofibrate (FF) did not significantly influence the studied end-points:

Figure S12 shows that FF-treatment did not affect the recruitment of neutrophils to the peritoneal cavity in the thioglycollate-peritonitis model 12 h post-injection.

Figure S13 shows that in DSS-induced colitis FF-treatment did not affect clinical signs of disease severity such as loss of body weight (A). Also, FF did not affect colon length (B), nor did it influence colonic expression of pro-inflammatory genes including IL-1β, IL-6 and TNFα (C-E), and did not significantly affect histological colitis scoring (F). Taken together, our data convincingly show that the effect of fibrates is restricted to gram-negative bacterial sepsis.

Figure S12**Figure S12. Fibrates do not affect neutrophil migration in thioglycollate-induced peritonitis.** Sterile inflammation was induced in C57BL/6 mice by i.p. injection of 3 ml 4%-thioglycollate. After 12 h, number of intraperitoneal neutrophils was determined with the XE 2100TM cell counter (n=5). Data are means ± SEM, n.s.= non-significant.**Figure S13. Fibrates do not influence the course of DSS-colitis.**

Male C57BL/6 mice treated with 0.2%-enriched FF or control chow were administered 3% DSS dissolved in water for 7 consecutive days. Thereafter, DSS was replaced by drinking water and all animals were followed up for another 7 days. (A) Changes in body weight (bw) and (B) colon length; (C-E) qRT-PCR analysis of immune response genes in colons of these mice, and (F) histopathological colitis scores with each point representing an individual mouse (n=8). Data are means ± SEM.

Figure S13

3) *How do fibrates affect polymicrobial sepsis, e. g. in a cecal ligation and puncture model?*

We fully agree with the Reviewer that it would be interesting to study the effect of fibrates on polymicrobial sepsis. However, we think that the two additional models requested by the Reviewer clearly show the mechanism of action of fibrates in the setting of bacterial sepsis. Additional studies with other infections e.g. with polymicrobial infection in the cecal ligation and puncture model are unlikely to add new information on the newly described mode of action of fibrates towards amelioration of sepsis and bacterial elimination by neutrophils. We agree that future studies with different other infection models and pathogens may be of interest to further explore the roles of fibrates in these different settings, however, they are beyond the scope of the current investigations identifying and analyzing a novel mechanism of fibrates in septicemia.

4) *Since fibrates have been around for a while, I am surprised that there is no clinical data on their effect in sepsis. The authors should dig deeper into the literature and maybe attempt to perform a meta-analysis of the published human clinical data.*

We agree with the Reviewer that the idea of analyzing the effect of fibrates on bacterial infection in humans is intriguing, and we believe that this should be done in a prospective randomized trial for the following reasons: Fibrates have been used to treat hyperlipidemia for more than 40 years. Six major studies (the CDP, WHO, HHS, VA-HIT, BIP, and FIELD studies) have been conducted to evaluate the safety and effectiveness of fibrates; however the results of these studies have produced mixed findings when evaluating overall mortality. The inconsistent outcomes may be a result of differences among individual fibrates and highly varied study populations (Backes et al. *Pharmacotherapy*, 2007 Mar;27(3):412-24). To our knowledge, there is no single trial on fibrate treatment which investigated the frequency of severe infections as a primary or secondary end-point. Moreover, even as a read-out of side effects the above mentioned studies did not give any indication

of the likelihood to develop a bacterial infection and/or severe sepsis under fibrate treatment. In fact these trials were designed to detect a reduction in coronary heart disease events in the intention-to-treat analysis, and neither bacterial infection nor sepsis were investigated end-points. In summary, we believe that only a prospective randomized trial will help to understand whether fibrates help to treat sepsis as adjunct immunomodulatory therapy in humans.

5) *How do fibrates affect sepsis survival in PPAR α knockout mice?*

We show in **Figure 3** of the manuscript that in *Ppara*^{-/-} mice fibrates exert effects on neutrophil recruitment and bacterial elimination fully resembling those observed in C57BL/6 wild-type mice, highly suggesting a similar beneficial effect on survival. As we thus did not expect additional significant information from survival studies in *Ppara*^{-/-} mice, and as we tried to limit the number of animals used in the study to a minimum in accordance with our animal research ethical approvals, we did not perform additional survival experiments in these specific knockout mice.

Referee #3 (Remarks):

In the present study Tancevski et. al. investigate the potential protective actions of fibrates in a murine model of bacterial induced sepsis. The authors provide evidence that administration of fenofibrate leads to protection from Salmonella typhimurium induced sepsis. They also demonstrate that PPAR α , the receptor described to mediate the cardiovascular actions of fibrates is not involved in this mechanism, whereas CXCR2 is key in mediating the protective actions of fenofibrate in vivo and WY-1463 in vitro. The experiments appear to be well conducted and the overall concept is very intriguing and could have potential clinical impacts. However there are several points that need to be addressed. They are as follows:

1) *The actions demonstrated for WY-1463 are in the **high micromolar range**, what are the **circulating levels of fenofibrate** that are reached via dietary supplementation. In addition how do these levels correlate with the some the **toxic effects associated with fibrates**?*

Fenofibrate and WY-14643 reportedly have high affinity to murine and human PPAR α . Fenofibric acid, the active compound of fenofibrate was shown to be highly specific for activation of PPAR α at concentrations of up to 100 μ mol/L. On the other hand, WY-14643 was found to selectively activate PPAR α at concentrations up to 30 μ mol/L. The C_{max} of 300 mg fenofibrate given orally is ~ 30 μ mol/L in humans (Arakawa et al. *ATVB*, 2005; 25: 1193-1197; Fruchart et al. *Curr Atheroscler Rep*, 2001; 3: 83-92). Thus, the dosages of fibric acids used in our *in vitro* studies not only served to selectively activate PPAR α , but are also in a similar molar range achieved in the *in vivo* setting in clinical routine in humans.

2) *The authors demonstrate that **PPAR α is not required** for the protective actions displayed by **fenofibrate in vivo**, is this true also for the actions demonstrate for WY-1463 in vitro? The authors also need to discuss the rationale for using of a different compound for their in vitro studies to that employed for the in vivo study.*

We did not test the effect of WY in PPAR α knockout cells *in vitro*; however, WY and fenofibrate are often combined in experimental studies to prove the class specificity for these drugs. One of the leading PPAR α researchers, Bart Staels, repeatedly used fenofibrate *in vivo* in mice and tested the drug class effect *in vitro* by using WY. To name one important study which also served as methodological reference paper for our present work is the study performed by Giulia Chinetti et al. published in *Circulation*. 2000; 101: 2411-2417, where the authors used a 0.2% fenofibrate enriched diet (wt/wt) *in vivo* in mice and 10 μ M WY *in vitro* in human and murine cells.

3) *The authors show that WY-1463 inhibits the down regulation of CXCR2 by LPS and preserves PMN chemotactic abilities. In addition they also demonstrate that in vivo fenofibrate increases early PMN recruitment to the site of inflammation and that inhibition of CXCR2 in vivo reverses these*

protective actions. Since PPAR α is not involved in these protective actions are these actions mediated by another receptor or do the authors propose that **these fibrates may directly regulate ERK and GRK2 phosphorylation?**

PPAR ligands were shown to exert so-called non-genomic effects, by inducing rapid intracellular MAPK-dependent signaling without PPAR activation. MAPKs including ERK1/2 and p38, in turn appear to be activated by fibrates such as WY through binding and induction of autophosphorylation of the epidermal growth factor receptor (EGFR), and alternatively by increasing the intracellular calcium levels. In addition, Src-dependent EGFR transactivation has been discussed. Please see the Review on this topic by Gardner, Dewar and Graves published in *Mol. Pharmacology* in 2005, 68:933-941.

According to the recent paper by Liu et al. (*Nat Immunology*, 2011;13: 457-464), phosphorylation of ERK and and/or p38 differentially affects neutrophil migration through rapid regulation of GRK2 expression, leading to modulation of the expression of cytokine receptors such as FPR1. Here, we show that such a mechanism takes place when neutrophils are treated with fibric acids, leading to a MAPK/GRK2-dependent promotion of chemotaxis via CXCR2, independently of FPR1 and LTB $_4$ R ligands.

4) It is note worthy that fenobirates also increase bacterial phagocytosis *in vivo*, do they also increase bacterial killing either by stimulating **ROS production** or by up regulating other **bactericidal mechanisms** such as bacterial peptides and NET formation. **AND**

6) On page 11 the authors state that 'This suggested that fibrates act on neutrophil chemotaxis mainly by influencing CXCR2, without affecting FPR1'. However no evidence is provided that FPR1 ligands are important in the *in vivo* experiments for recruitment of neutrophils to the site of inflammation. In addition the authors need to conduct **side-by-side experiments in vitro with both FPR1 agonists (such as fMLP) and CXCR2 agonists** to determine if in fact the reduction in receptor phosphorylation by GRK2 is restricted to CXCR2.

We thank the Reviewer for these suggestions. The newly performed experiments critically improved the quality of our manuscript by elucidating the effect of fibrates on neutrophil bacterial uptake, killing (ROS, MPO), and migration towards other ligands including fMLP and LTB $_4$. As shown below, and now implemented into the manuscript as supplementary files, fibrates did not affect the uptake of *S. typhimurium* into neutrophils (**Figure S5**). Regarding killing, we examined neutrophil ROS activity by performing a DCF assay and found that fibrates drastically decreased ROS formation (**Figure S6**). Moreover, fibrates markedly decreased degranulation of neutrophils, as shown in **Figure S7**. In addition, and following the suggestions of both Reviewers 1 and 2, a chemotaxis assay with neutrophils against different ligands including fMLP and LTB $_4$ was performed (**Figure S9**). In line with our hypothesis, both LPS as well as fibrates influenced only the migration towards the specific CXCR2 ligand CXCL2 (**Figure S9** and **Figure 4B**), while neither LPS nor fibrates affected the migratory properties of neutrophils towards fMLP and LTB $_4$. Taken together, our results show that fibrates promote the migratory efficiency of neutrophils in mice infected with *S. typhimurium*, leading to early clearance of infection without affecting their phagocytic properties, and that this effect critically depends on the regulation of CXCR2. Concomitantly, fibrates appear to dampen the killing activity of murine neutrophils, which *in vivo* may protect from an overwhelming inflammatory response and associated end-organ failure.

Figure S5

Figure S5. Fibrates do not affect neutrophil bacterial uptake.

Freshly isolated murine neutrophils were incubated with vehicle (DMSO) or the fibrate WY-14643 (WY, 10 μ M) for 30 min. Then neutrophils were infected with *S. typhimurium* at a MOI of 100 for 30 min at 37°C. Infected cells were washed with gentamicin to kill extracellular bacteria and harvested in 0.5% sodium deoxycholic acid. The lysates were plated onto LB agar plates and incubated at 37°C. CFUs were determined after 24 h. Data are means \pm SEM, n = 7.

Figure S6**Figure S6. Fibrates decrease neutrophil ROS activity.**

Reactive oxygen species (ROS) production was determined by flow cytometry, using 2', 7'- Dichlorofluorescein diacetate. Freshly isolated murine neutrophils were pre-stimulated with vehicle (DMSO) or with the fibrate WY (100 μ M) for 30 min at 37°C, and then incubated with heat-inactivated *S. typhimurium* at a MOI of 10 for 15 min. Results are expressed as mean fluorescence intensity, the histogram shows means for control and WY-treated cells, n = 5.

Figure S7**Figure S7. Fibrates decrease neutrophil MPO activity.**

Myeloperoxidase (MPO) activity was measured in freshly isolated murine neutrophils pre-stimulated with vehicle or with the fibrate WY (100 μ M) for 30 min at 37°C. Data are means \pm SEM, n = 6.

Figure S9**Figure S9. Fibrates selectively improve CXCR2-mediated chemotaxis of neutrophils.**

Chemotaxis to LTB₄ (100 nM), fMLP (100 nM) and CXCL2 (30 ng/ml) was determined in C57BL/6 bone marrow neutrophils pre-treated for 1 h with WY (10 μ M), or in combination with LPS (1 μ g/ml), as indicated. Data are pooled from 2 independent experiments.

In addition the authors need to provide detailed information on how the in vivo bacterial phagocytosis was evaluated in these experiments since no information is give in the manuscript in this regard.

We thank the Reviewer for this indication. We now added the requested information to the experimental procedures part of the manuscript.

5) *Is CXCR2 expression in vivo also preserved in mice treated with fenofibrates?*

As shown in **Figure S10**, fenofibrate treatment preserved CXCR2 expression on blood neutrophils measured as early as at 6 h post intraperitoneal infection with *S. typhimurium*, corroborating our *in vitro* findings with WY.

Figure S10**Figure S10. Fibrates inhibit the down regulation of CXCR2 on neutrophil *in vivo*.**

The expression of CXCR2 on neutrophils was determined by FACS analysis in whole blood of *Salmonella*-infected control or 0.2% FF-treated C57BL/6 mice 6 h after intraperitoneal inoculation with the pathogen. The histogram shows mean CXCR2 expression, $n = 5$.

7) What is the influence of fibrates on the **recruitment of other phagocytes to the site of inflammation such as monocytes?**

Figure R2
Interestingly, fibrate treatment increased the numbers of several leukocytes within the peritoneal cavity of *Salmonella* infected mice 12 h post-infection as measured using the XE 2100TM cell counter (**Figure R2**). The most pronounced increase was observed for neutrophils which, as major determinant of the early response in innate immunity may have consecutively attracted lymphocytes (Lym), monocytes (Mon), and eosinophils (Eos).

8) It is noteworthy that the message for a number of **Th1 cytokines, such as IFN γ** , which have been suggested to exert protective actions in sepsis are down regulated. Is this regulation observed at the **protein level too?** Also a discussion is merited for this apparent discrepancy between the current finding and those **published in the literature**.

Thank you for this comment. The reduction of cytokine expression in fibrate-treated mice experiencing *Salmonella* infection in comparison to infected littermate controls is a reflection of improved clearance of bacteria from the peritoneal cavity. This was further confirmed by additional experiments (**Figure S1**) showing that LPS injection into control and fibrate-treated mice did not result in changes of the mRNA expression of the different cytokines.

Figure R1
We definitively agree that IFN γ plays an important functional role in sepsis but its effects require further investigations. Data are partly contradictory and results may be affected by model-specific or species-specific variables that remain incompletely understood. In the CLP model of sepsis in the rat for instance, neutralization of IFN γ with a specific antibody resulted in a significant decrease of bacterial load in the peritoneum. Although an underlying mechanism could not be demonstrated, the authors speculated on reduced bacterial translocation following IFN γ neutralization (Qiu et al. 2001, Shock 16(6): 425-9). In contrast to these observations, IFN γ KO mice were found to be resistant to CLP-induced mortality (in the setting of antibiotic therapy). However, antibody-mediated neutralization of

IFN γ did not improve survival significantly. Thus, while IFN γ facilitated the pro-inflammatory response during CLP-induced septic shock, neutralization of IFN γ did not improve survival (Romero et al. J Leukoc Biol 88(4): 725-35). In the mouse model of peritonitis induced by colon ascendens stenting, lack of IFN γ in IFN γ KO mice was detrimental and associated with early mortality (Zantl et al. 1998, Infect Immun 66(5): 2300-9). Effects that could account for observed

differences but have not been evaluated in these studies are the effects of IFN γ on the expression of iNOS and the production of NO and its derivatives. NO may contribute to the mortality in sepsis by catecholamine-refractory hypotension secondary to vasodilation (Hollenberg et al. 2000, *Circ Res* 86(7): 774-8). A putative role of IFN γ and iNOS in the pathogenesis in pulmonary edema and ARDS, respectively, is even less clear (Heremans et al. 2000, *Am J Respir Crit Care Med* 161(1): 110-7). In humans however, it has long been appreciated that the strong primary immune response in sepsis may lead to exhaustion or deactivation of certain effector pathways so that host defense against microbes is subsequently undermined. IFN γ could counteract the phenomenon of immune deactivation in sepsis and improve monocyte function *in vitro* (Nalos et al., *Am J Respir Crit Care Med* 185(1): 110-2; Docke et al. 1997, *Nat Med* 3(6): 678-81). In our experiments in a microbial peritonitis model, we observed a significant down-regulation of splenic IFN γ mRNA expression (**Figure 1H** of the manuscript), while serum IFN γ protein levels were unaffected (**Figure R1**). This may result from differential IFN γ mRNA translation or IFN γ protein degradation (Ben-Asouli et al. 2002, *Cell* 108(2): 221-32). Our findings are compatible with the idea that a reduction of pro-inflammatory cytokines including IFN γ improves survival in Salmonella sepsis.

9) *The authors on page 5 state that spleen weight is a surrogate for the severity of infection, a reference for this statement should be provided.*

Several authors agree that increased spleen weight and disrupted spleen architecture is associated with increased disease severity in systemic Salmonella infection (Nairz et al. 2011, *Immunity* 34(1): 61-74; Mastroeni, et al. 2000, *J Exp Med* 192(2): 237-48; Monack et al. 2004, *J Exp Med* 199(2): 231-41). In line, a protective genotype is generally associated with reduced spleen weight and tissue damage (Brown et al., *Vet Pathol* 50(5): 867-76; Richer al., *Genes Immun* 12(7): 531-43; Zaki et al., *Proc Natl Acad Sci U S A* 111(1): 385-90).

10) *The authors employ IP injection of LPS to determine the actions of fenofibrate on splenic cytokine production, this approach assumes that bacteria do not escape the peritoneal cavity to directly elicit humoral responses in the vasculature and even in the spleen. Thus the authors should determine whether their results with i.p. administration of LPS also hold true when LPS is administered i.v.*

Intraperitoneal injection of sub-/lethal LPS doses is a commonly used model of systemic inflammation, critically affecting transcription of inflammatory genes in liver and spleen (please see a recent example: De Domenico et al. *J Clin Invest.* 2010;120(7):2395–2405). Moreover, a report by Jiang et al. clearly showed that LPS is measurable in plasma already 1 h after i.p. injection into mice (*Infect. Immun.* 1999, 67(4):1539). In addition, we did most experiments with viable bacteria which are then found in the circulation, thus resembling the situation of bacterial peritonitis and sepsis (please see also Nairz et al., *Immunity* 2011 and *J Exp. Med.* 2013).

11) *In a number of places in the results section (such as on page 7) the authors state that the mice were either treated with fenofibrate or solvent? What solvent does this statement refer to? In the materials and methods the authors state that the treatment was provided in the diet and not administered directly via injection.*

We apologize for this misleading formulation. As correctly noticed by the Reviewer, no solvent was used. Fenofibrate was directly mixed into the diet by the supplier (Ssniff Diets, Soest Germany). We changed the formulation into “control diet”.

Thank you for the submission of your revised manuscript to EMBO Molecular Medicine. We have now received the enclosed reports from the referees that were asked to re-assess it. As you will see the reviewers are now globally supportive and I am pleased to inform you that we will be able to accept your manuscript pending the following final amendments:

- Referee 2 has some final minor issues that i would like you to address in writing within the discussion section for point 1, and modify the corresponding sentences to points 2 and 3 as appropriate.

Please submit your revised manuscript within two weeks. I look forward to seeing a revised form of your manuscript as soon as possible.

***** Reviewer's comments *****

Referee #1 (Remarks):

The authors have performed a number of new experiments to address my prior concerns. Those studies support the original conclusion of the manuscript. Some of the experiments (e. g. the effect of fibrates on neutrophil respiratory burst and degranulation) also raise new questions, but the elucidation of the mechanism or relevance of those findings is beyond the scope of this already quite extensive study.

Referee #3 (Remarks):

The present manuscript is greatly improved,

There are a few minor points that need to be addressed:

Since intracellular ROS generation is critical for pathogen killing and therefore downregulation of ROS may be detrimental for host survival, the authors should distinguish between the reduction of extracellular ROS production, as measured in the present study, that may lead to tissue damage and intracellular ROS production.

The authors state " other ligands important for neutrophil recruitment, excluding the involvement of chemokine receptors other than CXCR2" Since statistical analysis was not conducted given that only 2 donors were employed and only 3 ligands were investigated the authors need to tone this statement down to reflect that results provided.

The authors state "Importantly, fibric acids not only reversed LPS-mediated CXCR2 degradation in vitro CXCR2 degradation in vitro", here the authors mean expression not degradation since they do not provide evidence for receptor degradation via proteolysis or proteasome breakdown,

2nd Revision - authors' response

07 March 2014

Specific points raised by the Reviewers:

Referee #1 (Remarks):

The authors have performed a number of new experiments to address my prior concerns. Those studies support the original conclusion of the manuscript. Some of the experiments (e. g. the effect of fibrates on neutrophil respiratory burst and degranulation) also raise new questions, but the elucidation of the mechanism or relevance of those findings is beyond the scope of this already quite extensive study.

Referee #3 (Remarks):

*The present manuscript is greatly improved,
There are a few minor points that need to be addressed:*

Since intracellular ROS generation is critical for pathogen killing and therefore down regulation of ROS may be detrimental for host survival, the authors should distinguish between the reduction of extracellular ROS production, as measured in the present study, that may lead to tissue damage and intracellular ROS production.

We thank for this comment. This point was addressed in the discussion part of our manuscript.

The authors state " other ligands important for neutrophil recruitment, excluding the involvement of chemokine receptors other than CXCR2" Since statistical analysis was not conducted given that only 2 donors were employed and only 3 ligands were investigated the authors need to tone this statement down to reflect that results provided.

We thank for this comment. This point was adequately addressed in the Results section on page 9 of our manuscript.

The authors state "Importantly, fibric acids not only reversed LPS-mediated CXCR2 degradation in vitro CXCR2 degradation in vitro", here the authors mean expression not degradation since they do not provide evidence for receptor degradation via proteolysis or proteasome breakdown.

We thank for this comment. This point was adequately addressed in the Results section on page 9 of our manuscript.